# Alterations of Amphetamine Reward by Prior Nicotine and Alcohol Treatment: The Role of Age and Dopamine

**DOI:** 10.3390/brainsci11040420

**Published:** 2021-03-26

**Authors:** Andrea Stojakovic, Syed Muzzammil Ahmad, Kabirullah Lutfy

**Affiliations:** 1Department of Pharmaceutical Sciences, College of Pharmacy, Western University of Health Sciences, 309 East 2nd Street, Pomona, CA 91766, USA; andrea.stojakovic@outlook.com (A.S.); smahmad@westernu.edu (S.M.A.); 2Department of Neurology, Mayo Clinic, 200 First St. SW, Rochester, MN 55905, USA

**Keywords:** amphetamine reward, nicotine, alcohol, conditioned place preference, dopamine transporter, age

## Abstract

Evidence suggests that nicotine and alcohol can each serve as a gateway drug. We determined whether prior nicotine and alcohol treatment would alter amphetamine reward. Also, we examined whether age and dopaminergic neurotransmission are important in this regard. Male and female adolescent and adult C57BL/6J mice were tested for baseline place preference. Mice then received six conditioning with saline/nicotine (0.25 mg/kg) twice daily, followed by six conditioning with saline/ethanol (2 g/kg). Control mice were conditioned with saline/saline throughout. Finally, mice were conditioned with amphetamine (3 mg/kg), once in the nicotine-alcohol-paired chamber, and tested for place preference 24 h later. The following day, mice were challenged with amphetamine (1 mg/kg) and tested for place preference under a drugged state. Mice were then immediately euthanized, their brain removed, and nucleus accumbens isolated and processed for the level of dopamine receptors and transporter and glutamate receptors. We observed a greater amphetamine-induced place preference in naïve adolescents than adult mice with no change in state-dependent place preference between the two age groups. In contrast, amphetamine induced a significant place preference in adult but not adolescent mice with prior nicotine-alcohol exposure under the drug-free state. The preference was significantly greater in adults than adolescents under the drugged state. The enhanced response was associated with higher dopamine-transporter and D_1_ but reduced D_2_ receptors’ expression in adult rather than adolescent mice, with no changes in glutamate receptors levels. These results suggest that prior nicotine and alcohol treatment differentially alters amphetamine reward in adult and adolescent mice. Alterations in dopaminergic neurotransmission may be involved in this phenotype.

## 1. Introduction

Tobacco smoking is a major public health issue and remains the single leading cause of preventable disease and death worldwide. Likewise, alcohol addiction is a significant public health and socioeconomic concern. The use of each drug alone, or in combination, is the main preventable cause of premature death worldwide, with an estimated death toll of about five million individuals annually. Notably, tobacco use can lead to nicotine addiction, and nicotine can serve as a gateway drug to facilitate alcohol intake and other addictive drugs [1,2,3,4,5,6,7,8,9,10,11,12,13,14,15,16,17,18]. In particular, nicotine has been reported to serve as a gateway drug for subsequent use and abuse of amphetamine, cocaine, and morphine [1,3,4,7,8,9,10,11,12,19,20,21,22,23,24,25].

Previous studies have shown that nicotine use is commenced during the adolescent period and manifests the use and abuse of alcohol and other drugs [2,26,27,28,29,30,31,32,33,34]. Previous studies have shown that nicotine exposure during adolescence alters the aversive [4] and rewarding [7,35] effects of cocaine. Additionally, it has been shown that aversion following nicotine withdrawal is reduced in adolescents than in adult rats [36], raising the possibility that adolescents are more prone to becoming polydrug users. Indeed, an earlier study has shown that the reinforcing action of methamphetamine is enhanced in adolescent rats with prior nicotine experience [37]. Therefore, in the present study, we determined whether the rewarding action of amphetamine would be altered by prior nicotine and alcohol exposure, and this response would be different in male and female adolescent mice (27–43 day-old) versus adult mice (60–120 day-old).

Dopamine plays a critical role in the locomotor stimulatory and rewarding effects of amphetamine in the dorsal and ventral striatum [38,39]. Previous studies have indicated the importance of dopamine transporter (DAT) in amphetamine reward [40]. Amphetamine has been shown to regulate the release of dopamine via the reversal of DAT. Amphetamine enters cells through DAT or by passive membrane transport. Once inside the presynaptic terminal, amphetamine inhibits monoamine oxidase. It also alters the vesicular storage of dopamine into synaptic vesicles. The accumulation of cytoplasmic dopamine leads to the reversal of DAT and dopamine transport to the extracellular space [41]. Therefore, the presence of functional DAT is necessary for the actions of amphetamine. Indeed, amphetamine-induced increases in accumbal dopamine and locomotion were reduced in DAT knockdown mice [42]. In contrast, mice overexpressing DAT displayed greater amphetamine-induced locomotor stimulation, extracellular dopamine, and reward [43].

The nucleus accumbens (NAc) is an integral brain region in the processing of reward. It receives dopaminergic input from the ventral tegmental area (VTA). The NAc receives glutamatergic input from the prefrontal cortex (PFC), amygdala, and hippocampus. Drugs of abuse, such as amphetamine, stimulate dopamine release in the NAc, leading to dopamine D_1_ or D_2_ receptors activation in the medium spiny neurons (MSNs). Therefore, we focused on the NAc and carried out Western Blot analyses to measure the level of DAT, dopamine D_1_ and D_2_, and glutamate receptors in this brain region.

## 2. Materials and Methods

### 2.1. Subjects

A total of 30 male and 29 female adolescent mice (27–43 days old) and adult mice (60–120 days old) on a C57BL/6 mouse strain were used throughout. Four litters were randomly assigned to the treatment groups, as described below. Mice were bred in house and maintained 2–4 per cage with free access to laboratory chow and tap water and kept under a 12 h light/12 h dark cycle in a temperature- and humidity-controlled room. The light was on at 6 a.m. and off at 6 p.m. All experiments were conducted during the light cycle between the hours of 10:00 a.m. to 5:00 p.m.; in accordance with the National Institute of Health for the Proper Care and Use of Animals in Research and approved by the Institutional Animal Care and Use Committee (protocol # R17IACUC018) at Western University of Health Sciences (Pomona, CA, USA).

### 2.2. Chemicals and Reagents

Ethyl alcohol (200 Proof), nicotine free base and amphetamine sulfate were purchased from OmniPur, EM Science (Gibbstown, NJ, USA), MP Biomedicals, Inc. (Solon, OH, USA), and Sigma/Aldrich (St. Louis, MO, USA), respectively. All drugs were dissolved in sterilized normal saline (0.9% sodium chloride solution in deionized water (St. Louis, MO, USA). Ethanol and amphetamine were injected intraperitoneally (i.p.) and nicotine subcutaneously (s.c.). The volume of injection was 10 mL/kg. Controls were injected with saline using the same volume of injection. Each drug solution was made daily before each conditioning and on the test day for state-dependent conditioned place preference (CPP).

### 2.3. Experimental Design and Procedures

#### 2.3.1. Place Conditioning Paradigm

The place conditioning paradigm, widely used as a measure of preference and aversion [44], was used to assess the motivational effects of nicotine and subsequent alcohol and amphetamine in adolescents and adult male and female mice. We used a 3-chambered place conditioning apparatus (ENV-3013, Med Associates Inc., St. Albans, VT, USA) with the smooth PVC floor in the central neutral chamber (9.78 × 12.7 × 12.7 cm) and two side chambers (16.76 × 12.7 × 12.7 cm) with wire mesh or grid rod floors as the tactile cues. The conditioning chambers were also decorated with black and white strips (2.54 cm), either vertically or horizontally, as the visual cues. There were sixteen holes in the sides for infrared beam strips. The chambers were covered with clear flip-top lids with ventilation holes. We used an unbiased and counterbalanced place conditioning paradigm described previously [45,46,47]. We first measured the baseline preference of each mouse toward the three chambers. The baseline measurements allowed us to assign equal numbers of mice to different tactile and visual cues and different treatments. Furthermore, this allowed us to establish a counterbalanced paradigm. On the baseline test day (day 1, D1), mice were placed in the central neutral grey chamber and allowed to explore the three chambers for 15 min. The amount of time that each mouse spent in each conditioning chamber was recorded. Mice then received twice daily (morning and afternoon) conditionings (15 min each) for three consecutive days (on days 2–4; D2–D4) with nicotine/saline or saline/nicotine (0.25 mg/kg; *n* = 7–8 mice of each age and sex group). The control groups received saline/saline (10 mL/kg; *n* = 7–8 mice of each age and sex group) conditionings and were then tested for place preference on day 5 (D5). The morning and afternoon conditionings were separated by four hours. On days 6–8 (D6–D8), mice received additional conditioning with nicotine (0.25 mg/kg) or saline and then were tested for CPP 24 h later (D9). Mice conditioned with nicotine then received three conditionings with ethanol (2 g/kg; 20% *v*/*v*) in the nicotine-paired chamber and saline in the saline-paired side and tested for CPP 24 h later (day 13, D13). On days 14–16 (D14–D16), mice received additional conditionings with ethanol or saline, as described above, and were tested for CPP 24 h later (day 17, D17). Saline-conditioned control mice were injected with saline instead of alcohol on the conditioning days and tested similarly. Finally, mice were conditioned (30 min each and once daily) with amphetamine (3 mg/kg) in the nicotine-alcohol-paired side and saline in the saline-paired chamber on days 18–19 (D18–D19) and tested for CPP 24 h later (D20). The following day (D21), mice were tested for a state-dependent CPP, i.e., under a drugged state, in which mice were challenged with amphetamine (1 mg/kg) and immediately tested for CPP. On each conditioning day, equal numbers of mice of different age groups were assigned to a drug or saline in the morning and the alternative treatment in the afternoon. On each post-conditioning test day, mice were placed in the central neutral chamber and allowed to explore the entire apparatus for 15 min. The amount of time that mice spent in each chamber was recorded in the same manner as for the preconditioning test day (Figure 1).

#### 2.3.2. Western Blot Analysis

On day 21, mice were then deeply anesthetized with isoflurane (32%) and decapitated. Brains were quickly removed, frozen on dry ice, and stored in a −80 °C freezer. The NAc was then dissected out and homogenized in RIPA lysis buffer (cat. # sc-24948, Santa Cruz, Dallas, TX, USA) containing phosphatase and proteinase inhibitors. Protein concentration was estimated by Pierce™ BCA Protein Assay Kit (cat. # 23225). Protein lysates (45 ng) were prepared in equal volume with 2× Laemmli Sample Buffer (cat. #1610737, Bio-Rad, Hercules, CA, USA) and loaded on 10% Mini-PROTEAN^®^ TGX Precast Gel (Bio-Rad). The gel was transferred to the PVDF membrane for 2 h at 200 mA. The membrane was then incubated with 1:1000 dilution of one of the primary antibodies, i.e., anti-mouse Actin, clone C4 (cat#MAB1501, Millipore, Burlington, MA, USA), anti-rabbit dopamine D_2_ receptor (cat# AB5084P, Millipore), anti-rabbit AMPA receptor (GluR1) (D4N9V) (cat#13185, Cell Signaling, Danvers, MA, USA), anti-rabbit NMDA receptor (GluN1) (D65B7) (cat# 5704, Cell Signaling); anti-rabbit dopamine transporter (cat# AB2231, Millipore); anti-rabbit dopamine D(1A) receptor (cat# ABN20, Millipore) overnight. The following day, the membranes were washed with tris-buffered saline (TBS) and 0.1% Tween 20 three times and exposed to a secondary antibody (1:1000 dilution of anti-rabbit/anti-mouse) at room temperature for one hour. The membrane was then washed three times, and Pierce™ ECL Western Blotting Substrate was added, and bands were exposed using a Bio-Rad ChemiDoc system. All antibodies were diluted in a 1:1000 ratio, using 5% milk or 5% BSA for phosphorylated proteins, made in tris-buffered saline (TBS) and 0.1% Tween 20.

### 2.4. Statistical Analyses

Behavioral data are presented as the mean (±S.E.M.) of the amount of time that mice spent in the drug-paired chamber (DPCh) and vehicle-paired chamber (VPCh), or the distance traveled (cm) in the DPCh and VPCh, and analyzed using a three-way analysis of variance (ANOVA). Western blot data were analyzed using a two-way ANOVA. The between factors were age and context, and time as the within factor for the place conditioning data. The Fisher’s LSD post hoc test was used to reveal significant differences between adolescent vs. adult mice. A value of *p* < 0.05 was considered significant.

## 3. Results

### 3.1. Amphetamine Induced a Comparable CPP Response under a Drugged State in Control Adolescent and Adult Mice

Figure 2 shows the amount of time that saline-treated control mice of each age group spent in the drug-paired (DPCh) and vehicle-paired (VPCh) chambers on days 1 (D1), 20 (D20), and 21 (D21). Three-way ANOVA revealed no significant effect of time (F_(2,81)_ = 1.21; *p* = 0.303), age (F_(1,81)_ = 0.16; *p* = 0.688), but there was a significant effect of context (F_(1,81)_ = 33.40; *p* < 0.0001) and a significant interaction between time and context (F_(2,81)_ = 13.94; *p* < 0.0001). The post hoc test revealed adolescent mice spent significantly more time in the amphetamine-paired than the saline-paired chamber. This result shows that these mice exhibited a significant CPP response after single-conditioning with amphetamine (Figure 2; the right half of the graph; compared DPCh vs. VPCh on D20). On the other hand, conditioning with the same dose of amphetamine failed to induce CPP in control adult mice (Figure 2; left half of the graph; compared DPCh versus VPCh). When these mice were challenged with amphetamine (1 mg/kg) and tested for CPP under a drugged state the following day (D21), both adolescent and adult mice exhibited a significant CPP response (Figure 2; compare DPCh versus VPCH on D21 in each group). This result suggests that adult and adolescent mice express a comparable state-dependent CPP. However, context after pairing with amphetamine may have gained more saliency in adolescent than adult mice.

### 3.2. Prior Nicotine and Alcohol Conditioning Increased the Rewarding Action of Acute Amphetamine under a Drugged State in Adult Compared to Adolescent Mice

Figure 3 illustrates the length of time that adolescent and adult mice with prior nicotine and alcohol exposure spent in the DPCh and VPCh before (D1) and after conditioning with amphetamine (D20), as well as on the test day for state-dependent CPP (D21). Three-way ANOVA showed a significant effect of time (F_(2,84)_ = 26.07; *p* < 0.0001), age (F_(1,84)_ = 5.76; *p* = 0 < 0.02), context (F_(1,84)_ = 61.64; *p* < 0.0001) and a significant interaction between time and context (F_(2,84)_ = 34.34; *p* < 0.0001), age and context (F_(1,84)_ = 14.30; *p* < 0.0003), and time, age and context (F_(2,84)_ = 5.93; *p* < 0.04). The post-hoc test revealed that single conditioning with amphetamine induced a significant CPP response in adult mice under a drug-free state (*p* = 0.01). On the other hand, conditioning with the same dose of amphetamine failed to induce CPP in adolescent mice with prior nicotine and alcohol experience (Figure 3). A trend toward a greater CPP response in adults than adolescent mice was observed following this single conditioning with amphetamine when animals were tested under a drug-free state (*p* = 0.08). The amphetamine challenge dose, given on day 21 (D21), induced a significant state-dependent CPP in mice of both groups, as evidenced by a significant increase in the amount of time that mice spent in the DPCh compared to VPCh (*p* < 0.001). This response was greater in adult than adolescent mice (*p* < 0.0001), showing that prior nicotine and alcohol experience increased amphetamine-induced state-dependent CPP in adult than adolescent mice.

### 3.3. The Expression of DAT and D_1_R Was Higher with a Concomitant Decrease in D_2_R in Adults Compared to Adolescent Mice with Prior Nicotine and Alcohol Exposure

Given that the action of amphetamine is mediated via the reversal of dopamine transporter (DAT) and the increase in the release of dopamine (DA) in the synaptic terminal, we hypothesized that nicotine and alcohol might differentially alter the expression of DAT and dopamine D_1_ (D_1_R) or D_2_ receptors (D_2_R) in adult compared to adolescent mice. There were two D_2_R bands. We only quantified the lower band and reported it here. Consistent with this hypothesis, DAT expression in NAc was higher in adult than adolescent mice with prior nicotine treatment (Figure 4). Additionally, the DAT level was significantly greater in treated adult mice than in other groups (*p* < 0.01). Furthermore, we observed a significant increase (*p* < 0.05) in the expression of accumbal D_1_R with a concomitant decrease in the level of D_2_R in adult mice compared to adolescent mice with prior nicotine and alcohol treatment. To assess whether these changes are selective to the dopaminergic neurotransmission, we also measured NMDA and AMPA expression (Figure 4). Our results showed there was no change in the level of AMPA and NMDA receptors between adult and adolescent mice with prior nicotine treatment.

## 4. Discussion

The current study provides evidence that control adolescent mice and adult mice exhibit a comparable state-dependent CPP. On the other hand, adult mice with prior nicotine and alcohol experience express a more robust amphetamine-induced state-dependent CPP than their adolescent counterparts, even though no significant CPP following nicotine or alcohol conditioning was found on days 5, 9, 13 or 17 in any age group (data not shown). Along with these behavioral changes, we observed increases in DAT and D_1_R and accompanied by a reduction in D_2_R expression in adult than adolescent mice with prior nicotine and alcohol experience. Together, these results suggest that prior nicotine and alcohol treatment differentially affected the rewarding action of amphetamine in adolescent and adult mice, with concomitant changes in the expression of proteins involved in dopaminergic neurotransmission.

Previous studies have shown differences in behavioral and neurochemical changes induced by amphetamines between adolescent and adult rodents [24,48,49]. Consistent with these earlier studies, we found behavioral changes between control adolescent and adult mice following a single amphetamine conditioning. Our results showed that amphetamine-induced CPP was greater under a drug-free state in adolescent mice than adult mice. On the other hand, treated adolescent mice showed a reduced CPP response following this single amphetamine conditioning. The reduced response in adolescent mice contrasts with previous studies showing that nicotine and alcohol can each serve as a gateway to heavier drugs [5,6,16,50,51,52]. However, consistent with the gateway theory, we observed a significant CPP after single conditioning with amphetamine in adult mice with prior nicotine and alcohol exposure (Figure 3) but not in saline-treated controls (Figure 3).

The gateway theory is referred to an increase in the use and abuse of heavier drugs following the initial use of nicotine or alcohol [5,6,16,50,51,52]. However, it may also mean that the use of nicotine preceding the use of other addictive drugs. In the present study, we observed a decrease rather than an increase in the rewarding action of amphetamine by prior nicotine and alcohol exposure in adolescent mice, which may contradict the gateway theory. However, one of the criteria for drug addiction is the development of tolerance to the pleasurable effect of addictive drugs on subsequent administration, leading to dose escalation [53]. If one applies the above theory of addiction, adolescent mice treated with nicotine and alcohol might need higher doses of the drug to achieve the same response. Therefore, they may be at a greater risk of developing addictive behaviors than adult mice. However, if one applies the positive reinforcement theory of addiction [54,55,56], the opposite would be true. Due to enhanced reward experience, adults most likely chase the drug and develop addictive behaviors faster than adolescents. This result may also apply to polydrug users, who may start using nicotine and alcohol during the adolescence period and then move on to heavier drugs, such as amphetamine. Therefore, further studies are needed to test these possibilities.

The CPP response under a drug-free state represents the motivational effect of the contextual cues that gain saliency following pairing with addictive drugs [45,57,58]. It appears that the context gains more motivational valence following amphetamine pairing in adolescents than adult mice, which may explain the vulnerability of this population to use drugs after subsequent exposure to the same environment, where they have used the drug once, i.e., the context may serve as a stronger stimulus in facilitating drug use in this population [24,48,49]. However, it appears that prior nicotine and alcohol exposure reduces this increase in the motivational valence of the context in adolescents but increases it in adult mice, as we observed a robust CPP in adult mice with prior nicotine and alcohol experience, but not in adolescent mice. The state-dependent CPP or the CPP response expressed under a drugged state, on the other hand, is a representative of the memory retrieval of the conditioned response that is acquired due to pairing of the subjective effects of the drug and the context during conditioning [45,57,58]. This response is more robust than the response under the drug-free state in adult compared to adolescent mice with prior nicotine-alcohol exposure. Although we expected to observe a greater response in adolescent mice, it appears that prior nicotine and alcohol conditioning may differentially bring about molecular changes, thereby leading to these behavioral changes.

Substances of abuse, such as nicotine, alcohol, and amphetamine, are thought to exert their rewarding effects through the activation of mesocorticolimbic dopaminergic neurons [59]. Previous studies have shown that the deletion of DAT expression in mice diminishes amphetamine-induced accumbal dopamine release and associated hyperlocomotion [42]. In contrast, mice with overexpression of DAT compared to their wild-type controls displayed greater locomotor activity in response to amphetamine, had higher extracellular dopamine, and showed a place preference to a lower amphetamine dose [43]. We observed a significant amphetamine-induced CPP in adult mice with prior nicotine and alcohol exposure under both drug-free and drugged states. This response was greater in adult than adolescent mice when animals were tested under the drugged state. On this basis, we hypothesized that prior nicotine and alcohol treatment differentially altered the expression of DAT or dopamine D_1_ or D_2_ receptors (D_1_R D_2_R) in adult mice compared to adolescents. Consistent with this hypothesis, we found that the expression of DAT and D_1_R increased in adult mice with prior nicotine treatment compared to their respective adolescent mice and control adult mice. There was a concomitant decrease in the expression of D_2_R in the NAc in adult mice with prior nicotine and alcohol experience compared to their respective adolescent mice. Considering that we used Western blot analysis, it is difficult to firmly state that the changes in D_2_R level are at the presynaptic or postsynaptic level. Therefore, future immunohistochemical studies are needed to shed light on this issue. However, it is noteworthy that these changes were specific to the dopamine system because we did not observe changes in the NMDA or AMPA receptors expression.

The observation that the expression of DAT and D_1_R increased and that of D_2_R decreased in adult mice with prior nicotine and alcohol exposure, compared to adolescent mice, may explain the underlying mechanism of the greater CPP response in adult mice compared to adolescent mice with prior nicotine and alcohol experience. We propose that the greater expression of DAT facilitates more amphetamine to gain access to the presynaptic terminal. Once inside the presynaptic terminal, it facilitates the reversal of DAT, allowing more dopamine to exit through DAT and reaching the postsynaptic D_1_R. Considering that there is more D_1_R expression, there is more opportunity for dopamine to bind to postsynaptic D_1_R and elicits a greater CPP response (Figure 5). Also, given that there is a decrease in D_2_R expression in these mice compared to adolescent mice with prior nicotine and alcohol exposure, we expect a reduction in binding of dopamine to D_2_R and thus a decline in a negative feedback mechanism, which also favors more dopaminergic neurotransmission to occur (Figure 5). However, further studies are needed to delineate whether these changes are due to nicotine or alcohol exposure or a combination of these drugs with amphetamine. Also, it needs to be verified whether the decrease in D_2_R expression is at the presynaptic or postsynaptic level.

Nevertheless, we do not believe that these alterations are due to amphetamine conditioning and amphetamine challenge alone. This notion is in line with the observation that we did not observe these molecular changes between control adult and adolescent mice. Also, we do not believe that these changes are due to only nicotine exposure because adolescent rats have been shown to have lower basal levels of dopamine compared to adults in tissue samples of the striatum [60], and reduced storage pool of releasable dopamine in this region [61]. However, baseline dopamine level in tissue samples of NAc and frontal cortex was reported to be comparable between adolescent and adult rats [60]. Furthermore, nicotine has been shown to increase dopamine release in the NAc in both adults and adolescent rats [62]. On the other hand, repeated ethanol administration in mice resulted in sensitization only in adults but not adolescents [63]. These authors also showed that adolescent mice exhibited lower dopamine levels in the PFC and NAc following ethanol exposure than adults [63]. Cumulatively, based on the published data, it could be suggested that a greater CPP response following amphetamine in adult mice, observed in our study, could be related to dynamic changes in an ethanol-induced decrease in dopamine release in NAc in response to alcohol conditioning between adolescent and adult mice or a combination of the action of nicotine and alcohol pre-exposure. We consider this as one of the caveats of the present study. Therefore, further studies are needed to assess the impact of preconditioning with each drug alone and their combination on the rewarding action of subsequent amphetamine treatment.

## 5. Conclusions

Our results showed that a single amphetamine conditioning induced a greater reward in control (no prior drug experience) adolescents than adult mice but induced a comparable state-dependent CPP in both age groups. On the other hand, the rewarding action of amphetamine under both drug-free and drugged-state was reduced in adolescents and enhanced in adult mice with prior nicotine and alcohol exposure. These changes were associated with increased DAT and D_1_ receptors with a concomitant reduction in D_2_ receptors expression. The decrease in the rewarding action of amphetamine in polydrug-exposed adolescents, compared with adult mice, may suggest that this may be one cause of the dose escalation in chasing pleasure in this vulnerable population.

## Figures and Tables

**Figure 1 brainsci-11-00420-f001:**
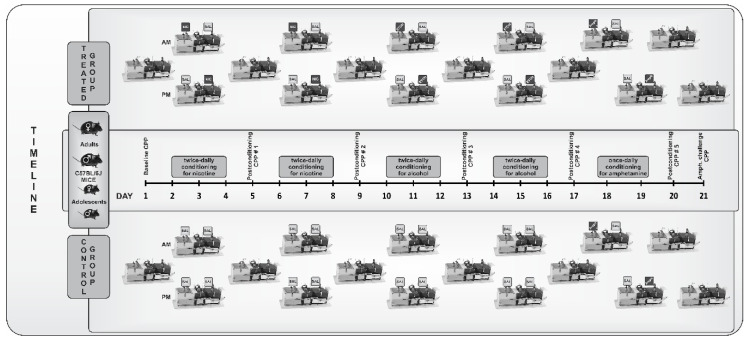
Schematic diagram of the place conditioning paradigm used in the current study.

**Figure 2 brainsci-11-00420-f002:**
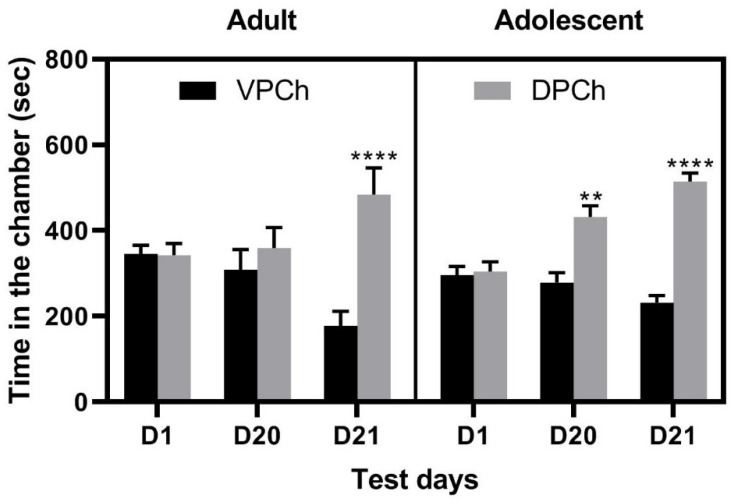
CPP induced by a single amphetamine conditioning in control adolescent and adult C57BL/6J mice. Male and female mice of both age groups (*n* = 7–8 mice per sex and age group) were tested for baseline and then given conditioning with saline in both place conditioning chambers. Mice were then conditioned with saline or amphetamine (3 mg/kg, i.p.) once daily in a counterbalanced manner and then tested for CPP 24 h after the last conditioning (D20). The following day, mice received a challenge dose of amphetamine (1 mg/kg) and tested for state-dependent CPP (D21). Data represent the amount of time that mice spent in the drug-paired chamber (DPCh) or vehicle-paired chamber (VPCh) on the baseline test day (D1), CPP test day under a drug-free state (D20), and drugged state (D21) and analyzed by three-way ANOVA followed by the Fisher LSD test. ** *p* < 0.01, **** *p* < 0.0001 vs. their respective VPCh.

**Figure 3 brainsci-11-00420-f003:**
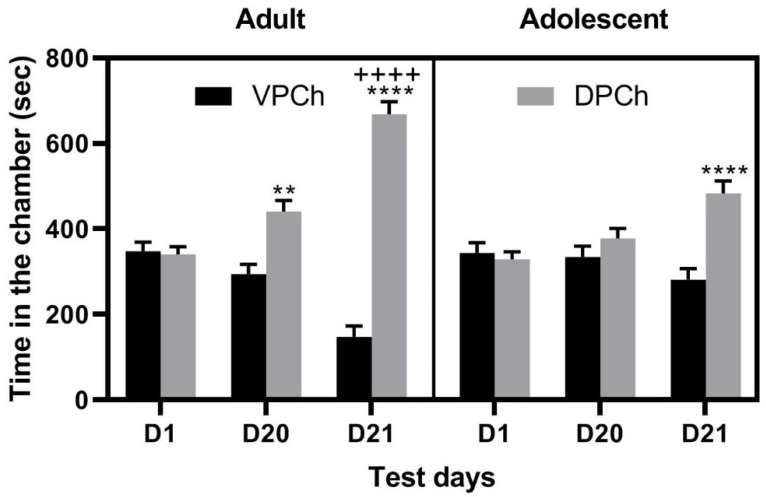
CPP induced by a single amphetamine conditioning in adolescent and adult C57BL/6J mice with prior nicotine and alcohol experience. Male and female mice of the two age groups (*n* = 7–8 mice per sex and age group) were tested for baseline and then received six conditioning with saline or nicotine (0.25 mg/kg, s.c.), followed by six conditioning with ethanol (2 g/kg, i.p.) in the nicotine-paired chamber and saline in the saline-paired chamber. Mice were then conditioned with saline or amphetamine (3 mg/kg, i.p.) once daily in a counterbalanced manner and then tested for CPP 24 h after the last conditioning. Amphetamine conditioning was carried out in the nicotine-alcohol-paired chamber. Data represent the amount of time that mice spent in the drug-paired chamber (DPCh) or vehicle-paired chamber (VPCh) on the baseline test day (D1), CPP test day under a drug-free state (D20), and drugged state (D21) and analyzed by three-way ANOVA followed by the Fisher LSD test. ** *p* < 0.01 versus its respective VPCh **** *p* < 0.0001 versus their respective VPCh; ++++ *p* < 0.0001 versus DPCh in Adolescents.

**Figure 4 brainsci-11-00420-f004:**
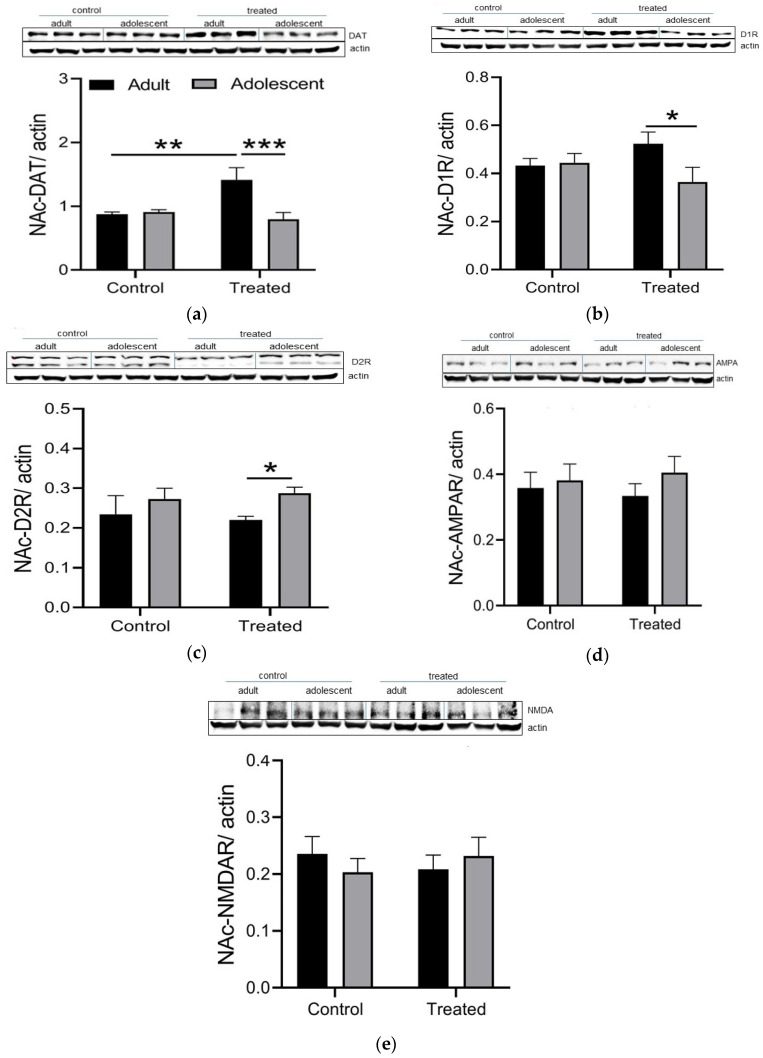
Expression of DAT and dopamine (D_1_ and D_2_) and glutamate (AMPA and NMDA) receptors in control and treated groups. Data represent the ratio of (**a**) dopamine transporter (DAT), (**b**) dopamine D1 receptors (D_1_R), (**c**) dopamine D2 receptors (D_2_R), (**d**) NMDA receptors, and (**e**) AMPA receptors over beta-actin as well as their corresponding representative bands from adult male and female mice and were analyzed by two-way ANOVA, and then subjected to between-group comparisons using the Fisher’s least significant difference test. (**a**) ** *p* < 0.01 vs. control group, *** *p* < 0.001 vs. treated adolescent group; (**b**) * *p* < 0.05 vs. treated adolescent group; (**c**) * *p* < 0.05 vs. treated adult group. Each band represents an individual mouse (8–10 samples per treatment/sex and each group).

**Figure 5 brainsci-11-00420-f005:**
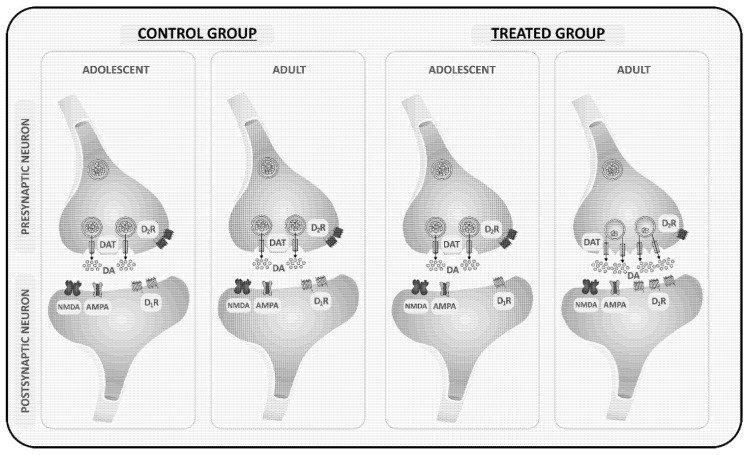
Schematic representation of changes in the level of DAT, D_1_ and D_2_ but not AMPA and NMDA receptors by prior nicotine and alcohol exposure, leading to altered amphetamine-induced stated-dependent CPP in adults versus adolescent mice. Amphetamine induced a comparable CPP under a drugged state in control adolescent and adult mice because the level of dopamine transporter (DAT) and dopamine (D_1_ and D_2_) receptors are not different between mice of the two age groups. On the other hand, the greater expression of DAT and D_1_ receptors and reduced levels of D_2_ receptors in adult than adolescent mice led to a greater CPP response under the drugged state in adult rather than adolescent mice with prior nicotine and alcohol experience.

## Data Availability

The data will be available upon request.

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
