# Peer review of "Alterations of Amphetamine Reward by Prior Nicotine and Alcohol Treatment: The Role of Age and Dopamine"

_brainsci, 2021, doi:10.3390/brainsci11040420_

Round 1
Reviewer 1 Report
The MS concerns interesting topic but needs revision.
1) The prior experience was both nicotine and ethanol, the authors should examine effect of each drug separately.
2)Sex difference should be analyzed.
3) The last sentence (lines 70-82) in Introduction should be remove.
Author Response
Review # 1
The MS concerns interesting topic but needs revision.
Response: Thank you for your consideration and valuable time to review our manuscript. We greatly appreciate your positive comments and insightful suggestions regarding our manuscript.
- The prior experience was both nicotine and ethanol, the authors should examine effect of each drug separately.
Response: This is an excellent point, and we agree with the reviewer that this issue needs to be addressed. However, due to COVID19, resources, and time constraints, it is not possible to conduct these studies. However, we included this in the Discussion as one of the caveats of the current study (please see the last paragraph of the Discussion).
- Sex difference should be analyzed.
Response: This is another important point. We agree with the reviewer, and we originally had planned to assess not only the impact of age but also sex. However, given we did not observe significant differences between male and female mice of either age group, we combined the data in male and female mice for each age group.
3) The last sentence (lines 70-82) in Introduction should be remove.
Response: As suggested, we removed the summary of our findings (Lines 70-82) from the Introduction.
Reviewer 2 Report
Good manuscript describing aterations of amphetamine reward in adult and adolescent mice.
Well designed and concluded.
Author Response
Review # 2 wrote:
Good manuscript describing alterations of amphetamine reward in adult and adolescent mice.
Well designed and concluded.
Response: Thank you for your consideration and valuable time to review our manuscript. We greatly appreciate the positive comments of the reviewer regarding our manuscript, and the design, and conclusion of our study.
Reviewer 3 Report
Stojakovic and colleagues investigated how a previous history of nicotine and alcohol would affect preference for amphetamine in a drug free as well as in an amphetamine priming protocol. They also correlated theit findings with molecular alterations in the dopaminergic system in the nucleus accumbens. the paper is relatively well written - however, I would ask the authors to clarify some issues that, in my opinion lowered the significance of the paper:
1) please add a timeline to the study as it is confusing to understand what happens in each day : test or conditionings; how many sessions per day, which drug ..
2) I couldn't understand why the authors did tests on D5 after half of the nicotine conditionings and also at D13 after hald of the ethanol conditionings. I can relate the tests performed on D9 and D17 after the nicotine and the alcohol conditioning.
3) it would be of interest to track the animals in a way to know is the animal that showed CPP to nicotine on D9 showed CPP to ethanol on D17 then CPP to amphetamine on D20 and 21 ? to have a clear picture of what is going on - would be of interest to show the data in a way naive vs animals with drug history with data from all conducted tests mainly the ones on D9 and D17 to track the history of these animals.
4) did the authors investigated any potential differences between males and females ?
5) how did the authors knew that the D2 receptors expression they are measuring by western blot are the presynaptic and not the postsynaptic ones as they are discussing their result in the discussion part and in Figure 4.
6) please keep in behavioural results figures the terms : adults and adolescents instead of old/young.
7) in the abstract line 24 - intro line 76-Figure 2 left side-discussion line 303-discussion line 350- conclusion line 356; the authors state that the rewarding action of amphetamine was increased in the adult group in a drug free state. However, when looking to the results line 201, they said that it was a tendency for significance but the P was equal to 0.08 meaning that no difference should be described regarding amphetamine reward for the adult group with a prior history of drugs in a drug free state. If I understood the authors correctly, I kindly ask them to adjust their conclusions and discussions with the observed non difference result.
8) abstract line 23, please add naive adolescent
9) results line 236, please add nicotine and alcohol treatment
10) discussion line 265, please reformulate and specify from which Figure this conclusion was drawn?
11) if I understood the authors correctly, the conclusion of Figure 2 is that adolescent and adult rats both express CPP to amphetamine when they received amphetamine priming before the test but the adults' preference was higher than the adolescents' one. so the *** in the adult left part at day 20 should be omitted if P=0.08.
12) the western blot of NMDA receptor band quality is very weak and difficult to quantify. for the D2 receptors expression, we see two bands, please specify which of these bands is the one corresponding to the size of the protein and is the one quantified.
Author Response
Stojakovic and colleagues investigated how a previous history of nicotine and alcohol would affect preference for amphetamine in a drug free as well as in an amphetamine priming protocol. They also correlated theit findings with molecular alterations in the dopaminergic system in the nucleus accumbens. the paper is relatively well written - however, I would ask the authors to clarify some issues that, in my opinion lowered the significance of the paper:
Response: We are grateful for your consideration and valuable time to review our manuscript. We also greatly appreciate your positive comments that “the paper is relatively well written” as well as your critical review of our manuscript and the insightful comments.
- please add a timeline to the study as it is confusing to understand what happens in each day : test or conditionings; how many sessions per day, which drug.
Response: We have provided a timeline for our experimental protocol (please see Diagram 1).
- I couldn't understand why the authors did tests on D5 after half of the nicotine conditionings and also at D13 after hald of the ethanol conditionings. I can relate the tests performed on D9 and D17 after the nicotine and the alcohol conditioning.
Response: In our earlier reports, we discovered that twice a day (once in the morning and once in the afternoon) saline/ethanol or ethanol/saline conditioning for 3-consecutive days induced a significant CPP in female mice (Nguyen et al., 2012; Tseng et al., 2013; Zaveri et al., 2018). We also found using the same place conditioning protocol with this dose of nicotine, we observed a significant CPP in female mice (Tseng et al., 2019). Therefore, we used the same protocol to assess if there is a difference in nicotine-induced CPP between adolescent and adult mice. However, we did not observe any CPP or CPA after this first set of conditioning (day 5), and therefore we continued them for three more days to see if longer conditioning would induce CPP. However, we did not see any CPP when mice were tested for CPP again on day 9. We then conditioned the mice with ethanol for three days as we expected to observe a CPP response based on our earlier studies (Nguyen et al., 2012; Tseng et al., 2013; Zaveri et al., 2018). However, we observed no CPP, and therefore we continued conditioning for three more days, and given we did not observe any CPP after each drug, we finally assessed if these treatments would alter amphetamine reward.
- it would be of interest to track the animals in a way to know is the animal that showed CPP to nicotine on D9 showed CPP to ethanol on D17 then CPP to amphetamine on D20 and 21 ? to have a clear picture of what is going on - would be of interest to show the data in a way naive vs animals with drug history with data from all conducted tests mainly the ones on D9 and D17 to track the history of these animals.
Response: This an excellent point, but given that we are given only seven days to submit the revised manuscript, it may not be possible to do this. However, if the reviewer believes that this is necessary, we will carry out such an analysis. If we find a clear picture, we will report it in the final version of the manuscript.
- did the authors investigated any potential differences between males and females ?
Response: As stated above, we originally planned to assess the influence of age and sex of mice. However, because we did not observe differences between male and female mice, we combined male and female mice for each age group.
- how did the authors knew that the D2 receptors expression they are measuring by western blot are the presynaptic and not the postsynaptic ones as they are discussing their result in the discussion part and in Figure 4.
Response: This is another excellent point. We agree with the reviewer that the Western Blot data cannot explain whether the decrease in the expression of D2 receptors is at the presynaptic or postsynaptic. We have included this as another caveat of the present study and proposed immunohistochemical studies as the future direction of the current project.
- please keep in behavioural results figures the terms : adults and adolescents instead of old/young.
Response: As recommended, we have removed the older figures 1 and 2 and replaced them with these changes incorporated in Figs 1 and 2.
- in the abstract line 24 - intro line 76-Figure 2 left side-discussion line 303-discussion line 350- conclusion line 356; the authors state that the rewarding action of amphetamine was increased in the adult group in a drug free state. However, when looking to the results line 201, they said that it was a tendency for significance but the P was equal to 0.08 meaning that no difference should be described regarding amphetamine reward for the adult group with a prior history of drugs in a drug free state. If I understood the authors correctly, I kindly ask them to adjust their conclusions and discussions with the observed non difference result.
Response: As suggested, we have revised these sentences to denote the difference was mainly under the drugged state, although a trend was also evident under the drug-free state between mice of the two age groups.
- abstract line 23, please add naive adolescent
Response: As suggested, we made this change.
- results line 236, please add nicotine and alcohol treatment
Response: As suggested, we made this change.
- discussion line 265, please reformulate and specify from which Figure this conclusion was drawn?
Response: As suggested, we made this change.
- if I understood the authors correctly, the conclusion of Figure 2 is that adolescent and adult rats both express CPP to amphetamine when they received amphetamine priming before the test but the adults' preference was higher than the adolescents' one. so the *** in the adult left part at day 20 should be omitted if P=0.08.
Response: The trend is between adult and adolescent mice on day 20 when the animals were tested under a drug-free state. We have modified this sentence to describe our results accurately.
12) the western blot of NMDA receptor band quality is very weak and difficult to quantify. for the D2 receptors expression, we see two bands, please specify which of these bands is the one corresponding to the size of the protein and is the one quantified.
Response: We apologize for the quality of these pictures. For the D2 receptor expression, we quantified the lower band. We included this information in the Result section (Please find this information under subheading # 3.3).
Round 2
Reviewer 3 Report
Thank you for ansewring my comments.
My answers are in the attachment highlighted in yellow.

Author Response
We thank the reviewer for his/her valuable time and consideration to further review our manuscript. We have responded to the constructive comments of the reviewer in the word document 11239947.v1.docx.
